# Socioeconomic Environment and Survival in Patients with Digestive Cancers: A French Population-Based Study

**DOI:** 10.3390/cancers13205156

**Published:** 2021-10-14

**Authors:** Laure Tron, Mathieu Fauvernier, Anne-Marie Bouvier, Michel Robaszkiewicz, Véronique Bouvier, Mélanie Cariou, Valérie Jooste, Olivier Dejardin, Laurent Remontet, Arnaud Alves, Florence Molinié, Guy Launoy

**Affiliations:** 1‘ANTICIPE’ U1086 INSERM-UCN, Normandie University UNICAEN, Centre François Baclesse, 14000 Caen, France; bouvier-v@chu-caen.fr (V.B.); olivier.dejardin@unicaen.fr (O.D.); alves-a@chu-caen.fr (A.A.); guy.launoy@unicaen.fr (G.L.); 2Service de Biostatistique–Bioinformatique, Pôle Santé Publique, Hospices Civils de Lyon, 69000 Lyon, France; mathieu.fauvernier@chu-lyon.fr (M.F.); laurent.remontet@chu-lyon.fr (L.R.); 3Laboratoire de Biométrie et Biologie Évolutive, Équipe Biostatistique-Santé, University of Lyon 1, CNRS, UMR 5558, 69100 Villeurbanne, France; 4Digestive Cancer Registry of Burgundy, Dijon University Hospital, INSERM UMR 1231, University of Burgundy, 21079 Dijon, France; anne-marie.bouvier@u-bourgogne.fr (A.-M.B.); vjooste@u-bourgogne.fr (V.J.); 5French Network of Cancer Registries, 31000 Toulouse, France; michel.robaszkiewicz@chu-brest.fr (M.R.); melan.daoulas@orange.fr (M.C.); florence.molinie@chu-nantes.fr (F.M.); 6Digestive Tumors Registry of Finistère, EA SPURBO 7479, CHRU Morvan, 29200 Brest, France; 7Digestive Cancer Registry of Calvados, Caen University Hospital, ‘ANTICIPE’ U1086 INSERM-UCN, Normandie University UNICAEN, Centre François Baclesse, 14000 Caen, France; 8Research Department, Caen University Hospital, ‘ANTICIPE’ U1086 INSERM-UCN, Normandie University UNICAEN, Centre François Baclesse, 14000 Caen, France; 9Department of Digestive Surgery, University Hospital of Caen, 14000 Caen, France; 10Loire-Atlantique/Vendée Cancer Registry, 44000 Nantes, France; 11CERPOP, Université de Toulouse, Inserm, UPS, 31000 Toulouse, France

**Keywords:** digestive cancers, cancer net survival, deprivation, social gradient, French cancer registries

## Abstract

**Simple Summary:**

Studies investigating the social gradient in digestive cancer survival are scarce, and the statistical methods used do not always consider important assumptions in survival analysis for adequate assessment. Using an ecological index (European Deprivation Index), we found a negative impact of social environment in digestive cancers net survival (especially for esophagus, stomach, bile ducts among females; colon and rectum for both sexes) and provided insight into how this social gradient in cancer survival builds up, and at what time of follow-up it appears. These results can guide clinical practice/public health actions to address social inequalities in survival by targeting digestive cancers with the greatest impact and identifying key follow-up periods to implement actions.

**Abstract:**

Social inequalities are an important prognostic factor in cancer survival, but little is known regarding digestive cancers specifically. We aimed to provide in-depth analysis of the contextual social disparities in net survival of patients with digestive cancer in France, using population-based data and relevant modeling. Digestive cancers (*n* = 54,507) diagnosed between 2006–2009, collected through the French network of cancer registries, were included (end of follow-up 30 June 2013). Social environment was assessed by the European Deprivation Index. Multidimensional penalized splines were used to model excess mortality hazard. We found that net survival was significantly worse for individuals living in a more deprived environment as compared to those living in a less deprived one for esophageal, liver, pancreatic, colon and rectal cancers, and for stomach and bile duct cancers among females. Excess mortality hazard was up to 57% higher among females living in the most deprived areas (vs. least deprived) at 1 year of follow-up for bile duct cancer, and up to 21% higher among males living in the most deprived areas (vs. least deprived) regarding colon cancer. To conclude, we provide a better understanding of how the (contextual) social gradient in survival is constructed, offering new perspectives for tackling social inequalities in digestive cancer survival.

## 1. Introduction

The role of social environment in health has been established for many years and concerns a wide variety of diseases, both chronic and acute, including digestive pathologies [1]. Research on social determinants of health rely on measures of social situation at the individual level (through indicators such as level of education, income, employment status etc.), or on contextual indexes that encompass more distal factors from the individuals (e.g., percentage of people below the poverty line, unemployed or low-skilled in a neighbor-hood, accessibility to services, medical premises, social network, etc.), providing a more comprehensive and integrated measure of their socio-economic environment and living conditions. Currently, beyond reporting, studies on the social determination of health are warranted to identify and understand the underlying mechanisms in an attempt to guide programs and practices aimed at tackling social inequalities [2]. To that end, and including in the field of cancer, studies based on unbiased general population data covering the entire social spectrum of patients would be helpful. Concerning the survival of patients with cancer in France, in agreement with the international literature based on either individual or contextual measures of social situation [3,4,5], a previous study showed that the prognosis was worse for the most disadvantaged populations than for the least disadvantaged ones, for most cancer sites, with a marked difference for some digestive ones, such as colorectal cancer and liver cancer in males or bile duct cancer and esophageal cancer in females [6].

However, in this previous study, as in the international literature on that subject, some elements that might help understand this social gradient were not fully explored. First, the key moments in the construction of these inequalities may differ according to the site, the mode of diagnosis, the availability of screening, the kind of treatment and other prognostic factors. Second, the gradient may not be expressed in the same way at all ages. Third, the pathway of social inequalities also depends on national contextual elements (e.g., public health policies) concerning the organization of primary prevention, screening and care. To date, much of the research on this topic has not analyzed net survival; thus, it has not been possible to distinguish between mortality due to cancer and that due to other comorbidities [7,8]. Moreover, as in the previous French study [6], among the studies based on the concept of cancer net survival, most used non-parametric analyses. Consequently, they did not account for baseline hazard flexibility and the putative time-dependent and non-linear effect of variables (i.e., social environment in our case) or interaction with age, which could be a limitation in cancer survival analysis [9]. It is possible that inequalities are built throughout the follow-up and add up through the different steps of cancer management (therapeutic choices, medical follow-up, treatment compliance, management of side-effects or relapses, etc.), leading to an increase in the social gradient of cancer survival over time. Conversely, it is possible that specific factors linked to the beginning of the cancer management induce social inequalities in cancer survival, which are no longer present thereafter, leading to a reduction in the social gradient of survival over time [10,11]. Furthermore, age-related factors could increase or reduce social inequalities in survival [12]. For instance, specific and close monitoring of patients in oncogeriatric departments could reduce the social gradient in this population. Conversely, the isolation or lack of autonomy of the elderly may make it worse.

The objective of this study was to provide in-depth analysis of the social disparities in survival at the contextual level in patients with digestive cancer in France for each cancer site, through flexible excess mortality hazard models using multidimensional penalized splines [13,14] and drawing on cancer registry population-based data.

## 2. Materials and Methods

### 2.1. Population and Data

The study population, which comprised 32,837 males and 21,670 females with diagnosed digestive cancer, was derived from the population-based data of three specialized digestive and 13 general French cancer registries belonging to the French Network of Cancer Registries (FRANCIM). All digestive cancer cases diagnosed and registered between 1 January 2006, and 31 December 2009 in patients over 15 years old were included, except for the Gironde and Lille area cancer registries for which cases were available only for 2008 and 2009, and for the Haute–Vienne cancer registry for which cases were available only for 2009. Cases were followed-up until the date of death or 30 June 2013 (except for loss to follow-up, which accounted for about 2% of all registered cases/cancers combined [6]). The study was approved by the Consultative Committee for the Processing of Health Research Data (CCTIRS) and the French Data Protection Authority (CNIL, authorization n° 913013).

Cancers were classified according to the topographical and morphological codes of the 3rd edition of the International Classification of Diseases for Oncology (ICD-O-3). Distribution of sites according to sex is shown in Table 1. Survival time corresponded to the difference between the date of death (or last information on vital status) and the date of cancer diagnosis. In about 0.1% of the cases for which the date of death was the same as the date of cancer diagnosis (survival time = 0), a survival time of 0.5 day was assigned. Other available data were age at diagnosis and social environment, assessed by the French version of the European Deprivation Index (EDI). For several years, there has been a wide consensus that socioeconomic status cannot be summed up by a single indicator. At individual level, socioeconomic status is usually explored in three fields: income, education and/or socio-professional category. The collection of individual socio-economic data consistently comes up against the problem of their absence from medical files or from medico-administrative databases (as in the French cancer registries), together with the issue of legal protection. Moreover, individual data may be exposed to a non-response bias in questionnaire surveys. In addition, assessment at individual level cannot account for contextual elements related to factors such as the place of residence (green spaces, criminality, equipment for physical activity, supply of consumer goods). Aggregated composite indexes have been constructed to allow the measurement of the social environment in large unbiased samples. Using a weighted combination of census data, they make it possible to integrate contextual elements and to assess the socio-economic environment at different area levels. For these reasons, we chose to use the EDI in this study, which is a European trans-cultural aggregate measure of relative poverty constructed with data from the European Union Statistics on Income and Living Conditions survey (EU-SILC) of EUROSTAT and declined in several European countries using census data [15]. It is currently available for seven European countries. The French version is available at the “IRIS” (Ilots Regroupés pour l’Information Statistique) level, which are the smallest geographic areas for which census data are available. There are more than 15.000 IRIS in France. The EDI was a continuous variable ranging from −16 to 56 (median value: −0.9) at the national level; the higher the index, the greater the deprivation in the IRIS. All the French IRIS are distributed into national quintiles of deprivation according to the EDI value nationwide, quintile 1 being the least deprived and quintile 5 the most deprived.

For each cancer case in the study population, the patient’s address at the time of diagnosis was geolocalized using Geographic Information Systems (ArcGIS 10.2, ESRI, Redlands, CA, USA) in order to be allocated to an IRIS with its corresponding EDI value and national quintile.

### 2.2. Statistical Analyses

All analyses were computed separately for each cancer site.

To model cancer-specific mortality in the absence of available data on the cause of death in the FRANCIM registries, analyses were conducted with the excess mortality framework [16]. Thus, at given values of time (t), age at diagnosis (a) and EDI, the observed mortality hazard h of an individual is as follows:(1)$$h\left(t,a,\mathrm{EDI},z\right)\;=h_{E}\left(t,a,\mathrm{EDI}\right)\;+\;h_{P}\left(a+t,z\right)$$
where $h_{E}$ is the excess mortality hazard (EMH), i.e., the mortality directly or indirectly due to cancer, and h_p_ is the expected mortality (h_p_ is the all-cause mortality hazard of the general French population at age a + t, given the demographic characteristics z of that individual). Here, z is composed of the variables sex, year of death and the residence Département (which is the main territorial and administrative division in France). The expected mortality h_p_ was provided by French life tables, produced by the National Institute of Statistics and Economic Studies (Institut National de la Statistique et des Etudes Economiques, INSEE).

The EMH was modeled using multidimensional penalized splines, which allows to model flexible baseline hazard, non-linear and non-proportional (i.e., time-dependent) effects of covariates as well as interactions [13,14]. This novel statistical model offers flexibility by using regression splines while limiting overfitting issues thanks to penalization. Four models based on penalized splines were adjusted and the best one was selected according to the corrected Akaike information criterion (AIC) [17]:

M0: $log\left(h_{E}\left(t,a\right)\right)\;=\mathrm{tensor}\left(t,a\right)$

M1: $log\left(h_{E}\left(t,a,EDI\right)\right)\;=\mathrm{tensor}\left(t,a\right)\;+\;s\left(\mathrm{EDI}\right)$

M1b: $log\left(h_{E}\left(t,a,EDI\right)\right)\;=\mathrm{tensor}\left(t,a\right)\;+\;s\left(\mathrm{EDI}\right)\;+\mathrm{tint}\left(t,\mathrm{EDI}\right)$

M2: $log\left(h_{E}\left(t,a,EDI\right)\right)\;=\mathrm{tensor}\left(t,a,\mathrm{EDI}\right)$

The keywords tensor, s, and tint respectively stand for a penalized tensor product spline, a one-dimensional penalized spline, and a penalized tensor product spline only containing interaction terms. Restricted cubic splines were used as one-dimensional splines or as marginal splines in a tensor product spline. We used 6, 5, and 5 knots for time, age, and EDI, respectively. The locations of these knots correspond to the percentiles of the distribution of time, age and EDI among deceased patients (this choice being justified by previous work [14]).

Smoothing parameters were estimated by optimizing the laplace approximate marginal likelihood (LAML) criterion and regression parameters by maximizing the penalized likelihood of the survival model. If M0 was selected, this meant that the effect of EDI on the EMH was considered as non-significant. If M1 was selected, the effect of EDI on the EMH was considered as significant and steady over time since diagnosis and identical, regardless of age at diagnosis. If M1b was selected, the effect of EDI was considered as significant and time-dependent but not age-dependent. If M2 was selected, the effect of EDI was considered as significant and age-dependent (or time- and age-dependent). The potential non-linearity of the effect of EDI (included as a continuous variable) was considered in all four models. The adequacy of the selected model was checked by comparing the net survival curves predicted by the model and those derived from a non-parametric method (Pohar-Perme) [7], using R software (R Core Team, Vienna, Austria, version 3.5.1) and the ‘relsurv’ (2.2.3) package.

Net survival probabilities and the EMH predicted by the selected model were then computed and plotted as a function of time since diagnosis, according to five key values for deprivation, defined as the median value of EDI in each quintile of the national distribution: mQ1 (least deprived, EDI = −4.2), mQ2 (EDI = −2.4), mQ3 (EDI = −0.9), mQ4 (EDI = 0.8), mQ5 (most deprived, EDI = 5.1). To represent the social gradient of cancer survival, the excess hazard ratio (EHR) of mQ5, mQ4, mQ3 and mQ2 versus mQ1 was computed. This was performed for several times of follow-up if the effect of EDI was found to be time-dependent, i.e., if M1b or M2 was selected.

Net survival methods assume that the death rate in the patient population is higher than the all-causes death rate in the background population. This is a reasonable assumption for cancers (especially digestive cancers), which is why such methods are relevant and commonly used in cancer studies. In addition, if this assumption would have been false, we would have encountered model convergence problems [7], which was not the case.

Since missing data for EDI accounted for less than 1%, we performed complete case analyses.

French life tables provided by INSEE are not stratified on deprivation, although background mortality in the general population might substantially differ according to socio-economic position; thus, social gradient in net survival for patients with cancer may be due at least partly to socially determined comorbidities. Therefore, as in previous studies [5,6,7,8,9,10,11,12,13,14,15,16,17,18], we conducted sensitivity analyses using two sets of simulated deprivation-specific French life tables. The simulations were based on the following: a) the mortality rate ratios by quintiles of the income domain score of the Index of Multiple Deprivation [19] provided by the deprivation-specific England life tables [20], England having large mortality inequalities as in France [21]; and b) the mortality rate ratios by quintiles of net income per consumption unit (individual level) provided by The Permanent Demographic Sample (Echantillon Démographique Permanent, EDP), a large-scale socio-demographic panel established in France [22]. Thus, in both scenarios, we applied the social gradient in mortality observed in the corresponding general population to the original French life tables. Since the external sources used for the simulations provided extreme social gradients in background mortality, our sensitivity analyses were conducted under “extreme correction” of the potential bias.

All the models were fitted using R software (3.5.1) with the “survPen” package (1.0.1) [23].

## 3. Results

Table 1 shows descriptive statistics by sex and cancer site as well as distribution of the study population into the national quintiles of deprivation and population net survival 1 month, 1 year and 5 years after cancer diagnosis provided by the best model selected by the AIC (see methods). Median age ranged between 66–77 years old across the cancer sites. As expected, 5-year cancer net survival probabilities were low for pancreas (males: 8.07%; females: 6.69%), liver (males: 14.61%; females: 14.22%), esophagus (males: 14.65%; females: 15.41%), bile ducts (males: 19.18%; females: 15.44%) and stomach (males: 23.7%; females: 27.69%) and higher for small intestines (males: 54.07%; females: 51.34%), rectum (males: 59.69%; females: 60.34%) and colon (males: 60.48%; females: 59.9%). Distribution of patients into the five national quintiles of EDI was around 20% for males, and it was a bit more heterogeneous among females, with less than 15% of patients in Q1 (least deprived) for esophagus or stomach, and 27.4% of patients in Q5 (most deprived) for liver cancer (resulting probably from a social gradient of incidence for these cancers).

As described in the Section 2, different models of the EMH were tested for each site and sex to assess whether net survival was influenced by EDI, and if so (M1, M1b or M2 model selected), whether this influence varied over time since diagnosis (M1b) and according to age at diagnosis (M2). As summarized in Table 2, net survival varied significantly according to EDI for all cancer sites but not for small intestine in both sexes (M0), nor for stomach and bile ducts in males (M0). It was dependent on time since diagnosis (M1b) of pancreas in males and for stomach, colon and bile ducts in females. This effect was not dependent on age at diagnosis for any site (no M2 selected).

Figure 1 shows the prediction of net survival by the selected model for each cancer site in the first five years after diagnosis for males (Figure 1a) and females (Figure 1b) according to medians of EDI national quintiles, when the selected model included an effect of EDI on net survival. Since the EDI effect was never dependent on age, we chose to represent net survival at 70 years of age at diagnosis.

Figure 1 shows how net survival regularly decreased with increasing deprivation score for each cancer site, reflecting the social gradient of survival. The gradient was particularly marked for colon, rectum, esophagus as well as bile ducts for females, with wider gaps between the five curves representing the five levels of EDI as compared to the other digestive cancers (pancreas, liver, as well as stomach among females).

Table 3 provides the estimates of excess mortality hazard ratios (EHR) between median values of national deprivation quintiles using mQ1 (least deprived) as reference for all cancer sites for which the EDI effect was significant. Detailed estimates are given at 1 month, 1 year and 5 years after cancer diagnosis for sites for which this effect was time-dependent.

Table 2 and Table 3 show that the effect of EDI was moderate for most digestive cancers studied. For most sites for which a significant effect of EDI on excess mortality was highlighted, the effect of EDI did not depend on time since diagnosis (4/5 sites for males: esophagus, colon, rectum and liver; and 4/7 for females: esophagus, rectum, liver and pancreas). For these sites, the prognosis progressively worsened with deprivation (Figure 1 and Table 3), with a regular and gradual worsening of the prognosis between the different quintiles of deprivation from the least deprived (first quintile) to the most deprived (fifth quintile). The pejorative effect of deprivation on net survival was particularly marked with esophageal (EHR_mQ5 vs. mQ1_: 1.44, 95% confidence interval (CI): 1.13–1.83), stomach (significant at 1 month of follow-up and not thereafter: EHR_mQ5 vs. mQ1 (1 month)_: 1.48, 95% CI: 1.09–2.03) and bile ducts (significant at 1 year of follow-up only: EHR_mQ5 vs. mQ1 (1 year)_: 1.57, 95% CI: 1.21–2.02) cancers among females, and with colon and rectal cancers in both males (colon, EHR_mQ5 vs. mQ1_: 1.21, 95% CI: 1.08–1.35; rectum, EHR_mQ5 vs. mQ1_: 1.2, 95% CI: 1.07–1.34) and females (colon, EHR_mQ5 vs. mQ1 (1 month)_: 1.2, 95% CI: 1.01–1.43 and EHR_mQ5 vs. mQ1 (1 year)_: 1.23, 95% CI: 1.06–1.43; rectum, EHR_mQ5 vs. mQ1_: 1.23, 95% CI: 1.09–1.39).

Table 3 and Figure 2 show the pattern of EDI effect changing over time for stomach, colon, and bile duct cancers in females and for pancreatic cancer in males. For stomach and colon among females and pancreas among males, the effect of deprivation on excess mortality was maximal during the first months or years after diagnosis, and then diminished over time, or even tended to reverse in the years furthest from diagnosis, although never significantly. However, for these three sites, the effect of EDI at the beginning of the follow-up was borderline significant, with the lower bound of the 95% CI close to 1.00. The effect of EDI was specific for bile duct cancer among females, with an increasing social gradient according to time since diagnosis. The effect was substantial as the EHR reached a value of 2.1 [1.1–3.9] from 3.8 years after diagnosis.

To assess the impact of potential bias associated with the lack of French life tables stratified on deprivation, two sets of simulated French deprivation-specific life tables were used. In these sensitivity analyses (Table 4), similar results were found overall, except for colon, rectal and esophageal cancers in males and colon cancer in females, for which no significant effect of EDI on net survival was highlighted by the selected model in the sensitivity analyses (Model M0 selected) contrary to main analyses (Model M1 or M1b selected). However, trends remained for these sites when M1 was tested in the sensitivity analyses, as was also the case in non-parametric analyses using the simulated life tables and Pohar-Perme method. In pancreas for males and bile ducts for females, the effect of EDI on net survival moved from time-dependent in the main analyses (Model M1b) to steady over time in the sensitivity analyses (Model M1). In all the other cases, results were steady from main analyses to sensitivity analyses (e.g., for liver cancer among females: EHR_mQ5 vs. mQ1 (main analyses)_: 1.18, 95% CI: 1.02–1.35; EHR_mQ5 vs. mQ1 (sensitivity analyses using England life tables)_: 1.17, 95% CI: 1.01–1.34; EHR_mQ5 vs. mQ1 (sensitivity analyses using EDP)_: 1.16, 95% CI: 1.01–1.34).

## 4. Discussion

Our results strongly suggest that there is a pejorative influence of deprivation (at the contextual level) on the prognosis of digestive cancers: esophageal, liver, pancreatic, colon and rectal cancers for both sexes, and stomach and bile duct cancers among females. The extent of the influence of social environment on net survival was greater for females than for males. In females, the maximum reached for bile duct cancer with an excess mortality hazard increased by 57% at one year of follow-up in the most deprived quintile compared to the least deprived quintile; in males, the maximum reached for colon cancer with an excess mortality hazard increased by 21% in the most deprived quintile compared to the least deprived quintile.

The use of net survival and flexible modeling of excess mortality due to cancer allowed us to show that the influence of deprivation on the excess mortality was similar in all age groups, that it could be time-dependent for some cancers, and that there was a progressive gradient across the social scale for all digestive cancer sites. The models showed that the social gradient of survival was observable from the first months or years after diagnosis for almost all digestive cancer sites, and that it remained throughout the patient’s care for most of them.

Social environment had a stronger effect on cancer survival in females. Except for esophageal and liver cancer, it is unlikely that this difference was due to differences in the biological or histological nature of the cancers. Similarly, as social environment was assessed in an aggregated manner using a geographical approach, it is unlikely that it was assessed differently for males and females. Therefore, these differences between males and females are likely due to the way in which cancers are diagnosed, managed and treated, as well as to a putative social determinism of participation in screening that is stronger in females than in males, particularly for colon cancer where these differences were marked. Unfortunately, due to the lack of data on the stage of extension at diagnosis or screening practice in our dataset, this hypothesis could not be tested.

Colon and rectal cancers are the cancers in which the impact of social environment on survival has been most studied, particularly in England. Our finding of an excess mortality risk greater than 20% for most deprived people as compared to least is consistent with published studies reporting social disparities in survival at the expense of the most deprived, whether it be colon cancer [4,24,25], rectal cancer [26,27] or colorectal cancer [18,28,29,30,31]. For colon cancer in females, our results suggest that social inequalities accumulate almost exclusively in the first months after diagnosis. This confirms data obtained with different models in England, Ireland and Spain, some of which explained social inequalities in survival mainly by the stage of extension at the time of diagnosis of the disease and treatment [24,27,30,32,33]. Similar results have been reported for rectal cancer with a high frequency of patients presenting in an emergency setting [27] and for both colon and rectal localizations combined [30]. However, other studies suggested that this gradient may develop at a distance from diagnosis, as suggested by the meta-analysis of Malietzis [34], which pointed out the relationship between social status and adjuvant chemotherapy modalities, and the study of Lyratzopoulos [26], which clearly showed that, before release, therapeutic innovations aggravate social inequalities in survival. Unfortunately, we could not investigate such a relationship because those data were unavailable.

Concerning liver cancer, our results show a significant effect of EDI on survival but with a smaller impact than for other digestive localizations, especially in males with an excess mortality risk of around 10% for the most deprived as compared to the least deprived. A pejorative and significant effect of social deprivation has been found in other studies conducted in the United States (SEER Program) [35,36], Korea [37], Australia and Canada [3,38], often with a stronger effect than in our study. The availability of powerful data from the CONCORD 3 study allowed Shao [39] to demonstrate a correlation between the Human Development Index, a composite measure of health, education, and economy, and liver cancer survival.

Similar studies are rarer for other digestive cancer sites. For pancreatic cancer as for other localizations, our results show that the influence of deprivation was maximal in the months following diagnosis, and that it was more marked in males than in females. These results are consistent with previous work that associated deprivation with less surgical resection [40,41,42,43,44].

We evidenced the social determination of survival even for esophageal cancer, which is characterized by a strong social determination of incidence [45] and short overall survival. In esophageal cancer, we found not only a stronger effect but also a clearer social gradient in females than in males, where there seemed to be a plateau effect for the most deprived groups. Our findings are consistent with previous research from Korea [37], the United States [46] and China [47].

Concerning stomach cancer, we did not find a significant effect in males, although the number of cases was equivalent to that of most of the other cancer sites. In females, the results are difficult to interpret with a strong but never significant reversal of the direction of the association between deprivation and excess mortality over the years after diagnosis. Our results are not consistent with those found in Korea in males over 60 years of age but are consistent with those found in England in 2008 [48,49]. Until now, we have been unable to explain the results concerning gastric cancer, in which there was a pattern of risk reversal over time for females. This issue therefore requires further investigation.

Apart from gastric cancer, the localizations for which there was no significant effect of social environment on survival were those with the lowest number of cases in our study. Concerning bile duct cancer in males, there was an excess of mortality that increased with deprivation but never significantly. In females, however, bile ducts were the localization where social deprivation had the greatest impact on survival, with a significant excess mortality risk of more than 50% one year after diagnosis for the most deprived females. A recent US study [42] using SEER data 2007–2015 reported the influence of social status on the surgical resection rate and survival, whereby the more privileged received wider resection.

We did not find any effect of social environment on survival for the small intestine in either sex. These results are consistent with those of the only other study we know of in small bowel cancer that did not evidence any effect of social deprivation on survival, even though the study included more than 5000 patients [50]. Similar results have been found regarding mortality inequalities among females [51].

In our study, a social gradient of survival was found for almost all digestive cancer sites, regardless of their prognosis, the availability of screening, the conditions of their diagnosis or their therapeutic management. The putative mechanisms underlying this phenomenon are numerous. Our results also suggest that there is a social gradient across the social spectrum, with a regular continuum from the most advantaged to the most disadvantaged, each social class having fewer chances of survival than the one immediately above. Similarly, we recently demonstrated how such a social gradient of survival is strong enough to create a social gradient of mortality, including for cancers such as colorectal cancer with the lowest incidence in the most deprived [52]. These findings rely on contextual/environmental social situation only since information at the individual level was not available in our data. Considering both levels and using multilevel analysis would have been more accurate and should be considered for future studies. Nevertheless, aggregated environmental indexes of deprivation have been recognized to be good proxies of the social situation at the individual level [53]. In addition, previous studies have shown that social environment itself may play a role in health related outcomes, especially cancer survival and incidence [54,55]. Our results therefore confirm these previous findings and underline the interest of also investigating the social context in which individuals live, in order to better understand the social determinants of cancer survival.

Our original statistical modeling methods revealing interactions over time showed that the social gradient of survival was not formed exclusively at a distance from diagnosis in any type of digestive cancer. For most sites, the absence of variation in excess mortality over time suggests that the construction of social inequalities occurs throughout the medical course of the disease, thus highlighting the role of the organization of care. However, for several sites, these inequalities are most likely to develop during the first few months following diagnosis. This phenomenon was particularly marked for colorectal cancer, thus highlighting the importance of access to screening in the development of social inequalities in survival [24,30].

Our study has several strengths. First, most studies that have examined this topic classically analyze crude survival with the Cox model. Studies similar to ours that model net survival [3,18,30,35,56] are free of gender- and age-related co-morbidities and can thus model excess mortality directly due to disease. Second, compared to the non-parametric evaluations of net survival, our flexible method allowed an in-depth population-based analysis and may have contributed to uncovering potential underlying mechanisms such as non-proportional and time-dependent effects.

The study also has limitations. First, the analysis was limited by the lack of data on cancer extension and modalities of treatment, which are the most important cancer prognostic factors, often related to social situation themselves. Unfortunately, such parameters are not routinely collected by the French cancer registries (which conversely present the advantage of providing exhaustive and high quality data with large coverage of the French population). A perspective to continue and complete this work would be to conduct a “high resolution” study with collection of various clinical and biological parameters, based on a smaller sample. Nevertheless, we think that our study provides a first highlight of the problem of social inequalities in digestive cancers survival in France and paves the way for future research. Second, in the absence of a mortality table of the general population as a function of the level of social deprivation, models such as ours do not allow socially determined causes of death to be considered and cannot be used to unequivocally attribute differences to a social differentiation in disease management, no matter how sophisticated they are. However, the sensitivity analyses, albeit conducted under a high hypothesis of error and revealing the presence of the expected overestimation bias, did not question the overall findings of the study. EHR reported in this study are probably overestimated due to that limitation, but sensitivity analyses suggest that the error might not be important enough to substantially modify the value of the EHR and to contradict our findings. Although these were only sensitivity analyses, to us, this method, which has already been used and approved in previous published studies [5,8,57], was the best way to confirm our findings. Nonetheless, we are aware that this bias will be correctly accounted for only when French life tables stratified by deprivation for the general population become available. Third, we may have introduced uncertainty because of multiple testing, and by using a two steps procedure to estimate excess hazard rates (i.e., choice of model using AIC and then prediction of excess hazard/net survival from this model) which is a common problem in studies on cancer net survival based on flexible model. However, use of multidimensional penalized splines considerably reduced the number of steps in the model building strategy by reducing the number of candidate models, thus limiting the number of tests and the uncertainty as compared to previous strategies. In the end, the extent of model uncertainty due to such an AIC strategy in a penalized setting, based on only four candidate models, is probably slight.

Considering these limitations, estimates provided in this study must be interpreted with caution, keeping in mind that the social gradient might be slightly overestimated.

From a methodological point of view, when required variables are available, the most relevant studies to analyze the underlying mechanisms of construction of social inequalities in health are those based on a mediation model, since they aim to quantify the relative contribution of the different paths of construction of these inequalities. The mediation analysis conducted by Frederiksen [29] is particularly interesting. After accounting for the potential effects of the stage of extension at diagnosis and the mode of treatment, it raises the possibility of a direct effect (i.e., not mediated by differences in cancer diagnosis or management) of the social environment on an individual’s ability to survive cancer. The highlighting of such a direct effect can provide support for the hypothesis of allostatic load [58], based on the exhaustion of stress defense mechanisms. The assumption is that the lower the financial, social or cultural capital, the greater the individual’s load. When repeatedly placed in conditions of difficulty to meet essential needs, this permanent state of demand (“social stress”) overstretches an individual’s ability to adapt, particularly their ability to cope in such a way that the necessary balance in life is ensured. When external demand exceeds an individual’s capacity to adapt, the mobilization of mechanisms to maintain the balance can become deleterious from a health perspective. Such a hypothesis has been confirmed in the field of cardiovascular and neurological diseases but has received less attention in the field of cancerology. Beyond the repeated observation of social inequalities in survival for patients with cancer in different national contexts, this hypothesis is one of the avenues that requires further exploration. In addition, the dataset supporting our findings is ancient and it would be interesting to reproduce this study with more recent data (when available), which will also enable us to compare periods and investigate evolutions in the social gradient in digestive cancers survival in France.

## 5. Conclusions

These research perspectives would be interesting to explore for digestive cancers, especially since our results confirm the existence of a systematic and unidirectional social gradient in survival for the majority of digestive cancer sites in France, and since it has been shown that the social gradient in survival contributes more strongly than the social gradient in incidence to the overall social gradient in cancer-related mortality [52].

## Figures and Tables

**Figure 1 cancers-13-05156-f001:**
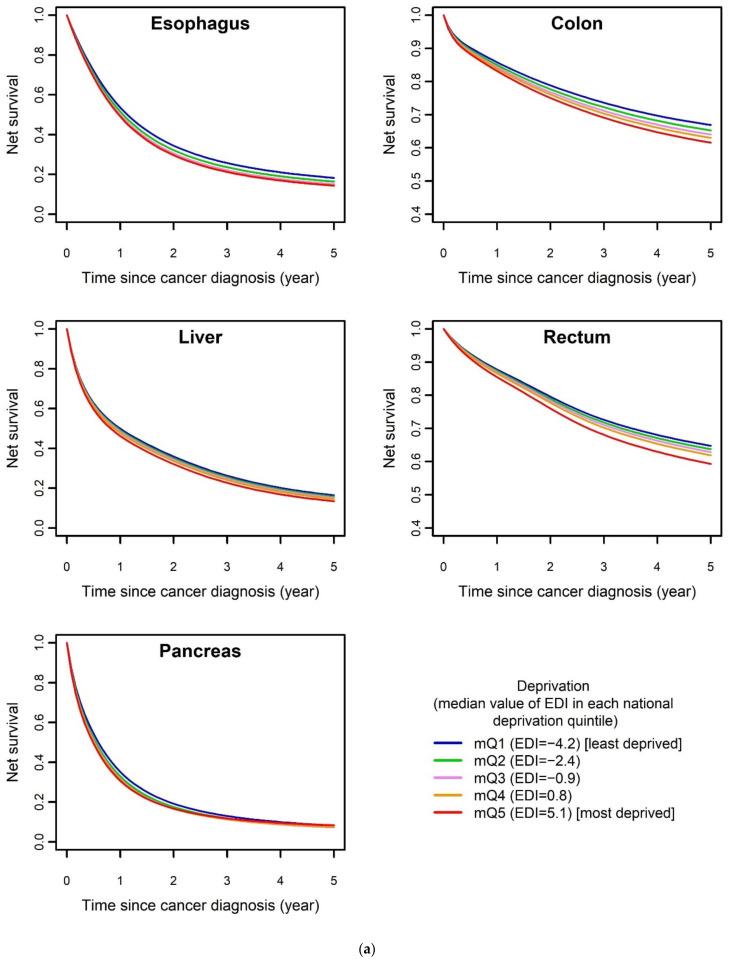
Net survival over time since diagnosis by level of deprivation for each cancer site for which EDI effect was significant, predicted by selected model for 70-year-old (**a**) males and (**b**) females. EDI: European Deprivation Index; *mQi*: median value of national deprivation quintile *i* (see Section 2).

**Figure 2 cancers-13-05156-f002:**
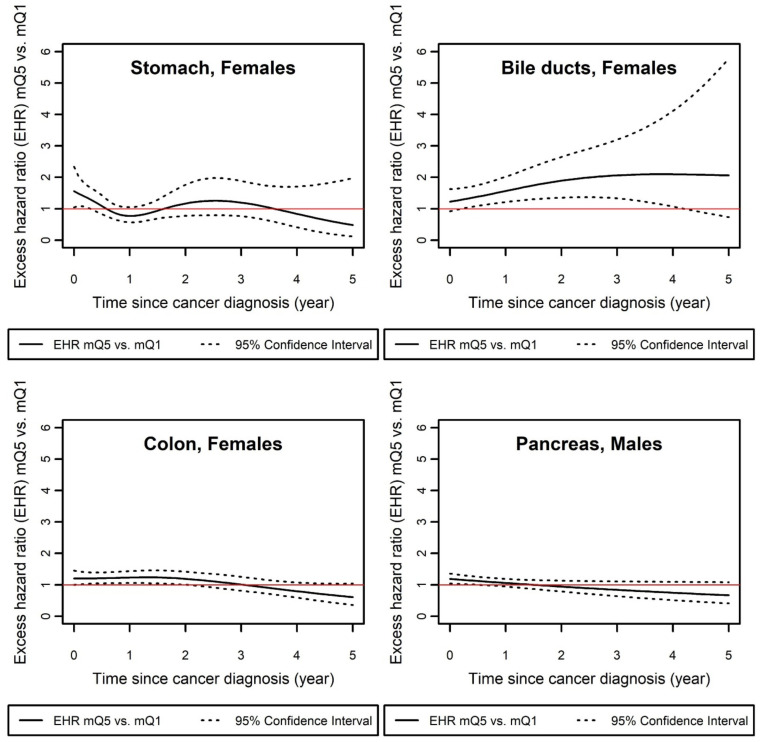
Excess hazard mortality ratio (EHR) between mQ5 (most deprived) and mQ1 (least deprived), over time for cancer sites for which effect of EDI on net survival was time-dependent (M1b selected). EHR: excess mortality hazard ratios; mQ*i*: median value of national deprivation quintile *i* (see Section 2).

**Table 1 cancers-13-05156-t001:** Sociodemographic characteristics, EDI national quintile distribution and populational net survival (computed from the selected model, at 5 years of follow-up) according to cancer site.

Cancer Site	Topography Code	*n*	*n* Deaths	Median Age	Q1 EDI (%)	Q2 EDI (%)	Q3 EDI (%)	Q4 EDI (%)	Q5 EDI *(%)*	5-Year Net Survival [95%CI]
** *Males* **										
Esophagus	C15	3250	2831	66	17.3	20.2	20.8	19.9	21.8	14.65 [11.98;17.66]
Stomach	C16	3493	2777	72	18.1	20.6	19.8	19.4	22.1	23.70 [20.89;26.66]
Small intestine	C17	512	261	68	19.7	21.5	20.3	20.7	17.8	54.07 [46.62;60.94]
Colon	C18	10,119	4815	72	18.6	21.3	20.9	20.0	19.2	60.48 [57.97;62.9]
Rectum	C19-C21	6220	2917	70	18.7	21.4	22.7	18.8	18.4	59.69 [56.69;62.57]
Liver	C22	4979	4308	69	19.2	20.7	19.4	20.2	20.5	14.61 [12.52;16.91]
Bile ducts	C23–24	848	710	73	20.0	17.2	19.7	21.3	21.7	19.18 [15.01;23.80]
Pancreas	C25	3416	3155	69	19.3	20.1	21.2	18.7	20.7	8.07 [6.06;10.5]
** *Females* **										
Esophagus	C15	705	599	72	14.8	19.9	19.4	24.4	21.6	15.41 [10.37;21.61]
Stomach	C16	1905	1407	77	14.9	19.7	20.2	21.7	23.6	27.69 [23.09;32.62]
Small intestine cancer	C17	399	201	70	18.5	20.1	19.0	19.0	23.3	51.34 [43.78;58.56]
Colon	C18	8669	4037	75	17.2	20.1	21.0	20.1	21.6	59.9 [57.24;62.43]
Rectum	C19-C21	4679	2064	72	17.6	21.0	19.9	20.4	21.1	60.34 [57.05;63.50]
Liver	C22	1115	956	74	15.1	18.4	19.6	19.5	27.4	14.22 [10.73;18.3]
Bile ducts	C23–24	1011	873	77	15.3	20.3	18.4	23.7	22.3	15.44 [11.55;20.01]
Pancreas	C25	3187	2962	75	15.9	19.2	20.6	21.4	22.8	6.69 [5.01;8.72]

EDI: European Deprivation Index; Q*i* EDI (%): proportion of individuals in population study belonging to national deprivation quintile *i*; 95%CI: 95% confidence interval.

**Table 2 cancers-13-05156-t002:** Effect of deprivation assessed by EDI on net survival according to cancer site and sex, as assessed by selected flexible model.

Cancer Site	Significant Effect of EDI	Effect of EDI Time-Dependent †	Effect of EDI Age-Dependent †	Model Selected ‡
** *Males* **				
Esophagus	YES	NO	NO	M1
Stomach	NO	---	---	M0
Small Intestine	NO	---	---	M0
Colon	YES	NO	NO	M1
Rectum	YES	NO	NO	M1
Liver	YES	NO	NO	M1
Bile ducts	NO	---	---	M0
Pancreas	YES	YES	NO	M1b
** *Females* **				
Esophagus	YES	NO	NO	M1
Stomach	YES	YES	NO	M1b
Small Intestine	NO	---	---	M0
Colon	YES	YES	NO	M1b
Rectum	YES	NO	NO	M1
Liver	YES	NO	NO	M1
Bile ducts	YES	YES	NO	M1b
Pancreas	YES	NO	NO	M1

EDI: European Deprivation Index; †: not applicable (---) if EDI effect was not significant; ‡: effect of EDI on excess mortality hazard: M0: not significant, M1: significant, steady over time since diagnosis and identical regardless of age at diagnosis, M1b: significant, time-dependent but not age-dependent.

**Table 3 cancers-13-05156-t003:** Excess mortality hazard ratio (EHR) estimates for median value of national deprivation quintiles according to sex and cancer site.

Cancer Site	Time of Follow-Up	EHR	EHR [95% CI]	EHR [95% CI]	EHR [95% CI]	EHR [95% CI]
Cancer Site †	Time of Follow-Up ‡	mQ1 (ref)	mQ2	mQ3	mQ4	mQ5
** *Males* **						
Esophagus	NA	1	1.06 [1.01–1.12]	1.11 [1.02–1.21]	1.14 [1.03–1.25]	1.14 [1.01–1.29]
Colon	NA	1	1.06 [1.01–1.11]	1.11 [1.03–1.19]	1.15 [1.06–1.25]	1.21 [1.08–1.35]
Rectum	NA	1	1.04 [1.01–1.06]	1.07 [1.03–1.11]	1.1 [1.04–1.17]	1.2 [1.07–1.34]
Liver	NA	1	1.02 [1–1.05]	1.04 [1–1.09]	1.06 [1.01–1.13]	1.11 [1.02–1.21]
Pancreas	1 month	1	1.07 [1.02–1.13]	1.11 [1.03–1.21]	1.14 [1.03–1.25]	1.18 [1.03–1.34]
	1 year	1	1.05 [1–1.1]	1.07 [0.99–1.16]	1.07 [0.98–1.17]	1.06 [0.94–1.19]
	5 years	1	0.96 [0.87–1.06]	0.91 [0.76–1.09]	0.84 [0.64–1.09]	0.67 [0.41–1.08]
** *Females* **						
Esophagus	NA	1	1.09 [1.01–1.18]	1.17 [1.03–1.33]	1.25 [1.06–1.48]	1.44 [1.13–1.83]
Stomach	1 month	1	1.18 [1.02–1.36]	1.33 [1.06–1.66]	1.43 [1.1–1.86]	1.48 [1.09–2.03]
	1 year	1	0.94 [0.83–1.06]	0.8 [0.65–0.99]	0.73 [0.56–0.95]	0.77 [0.57–1.05]
	5 years	1	0.89 [0.63–1.25]	0.79 [0.44–1.41]	0.68 [0.3–1.54]	0.49 [0.12–1.97]
Colon	1 month	1	1.09 [1.01–1.17]	1.15 [1.03–1.29]	1.18 [1.03–1.35]	1.2 [1.01–1.43]
	1 year	1	1.06 [1–1.11]	1.11 [1.01–1.22]	1.15 [1.03–1.3]	1.23 [1.06–1.43]
	5-year	1	0.9 [0.81–1]	0.83 [0.68–1]	0.75 [0.56–1]	0.61 [0.36–1.04]
Rectum	NA	1	1.04 [1.02–1.07]	1.08 [1.03–1.12]	1.12 [1.05–1.19]	1.23 [1.09–1.39]
Liver	NA	1	1.03 [1–1.06]	1.06 [1.01–1.11]	1.09 [1.01–1.18]	1.18 [1.02–1.35]
Bile ducts	1 month	1	1.05 [0.95–1.16]	1.1 [0.94–1.28]	1.16 [0.95–1.4]	1.25 [0.95–1.63]
	1 year	1	1.09 [0.99–1.21]	1.19 [1.02–1.38]	1.31 [1.08–1.58]	1.57 [1.21–2.02]
	5 years	1	1.15 [0.93–1.43]	1.31 [0.89–1.93]	1.51 [0.86–2.68]	2.06 [0.73–5.78]
Pancreas	NA	1	1.02 [1–1.03]	1.03 [1–1.06]	1.04 [0.99–1.1]	1.08 [0.99–1.19]

CI: confidence interval; EHR: excess mortality hazard ratios; mQ*i*: median value of national deprivation quintile *i* (see Section 2); NA: not applicable. † Except sites for which no significant effect of EDI on net survival was highlighted (M0 model selected); ‡ NA if no time-dependent effect of EDI was highlighted (M1 model selected).

**Table 4 cancers-13-05156-t004:** Selected model and excess hazard mortality ratio (EHR) between mQ5 (most deprived) and mQ1 (least deprived) according to analysis (main or sensitivity) for each cancer site and sex.

Cancer Site	Main Analyses			SA (1)			SA (2)		
	Selected Model †	Time of Follow Up	EHR mQ5/mQ1 [95% CI]	Selected Model †	Time of Follow Up	EHR mQ5/mQ1 [95% CI]	Selected Model †	Time of Follow Up	EHR mQ5/mQ1 [95% CI]
** *Males* **									
Stomach	M0	NA	NA	M0	NA	NA	M0	NA	NA
Liver	M1	NA	1.11 [1.02;1.21]	M1	NA	1.07 [0.99;1.16]	M1	NA	1.07 [0.99;1.15]
Esophagus	M1	NA	1.14 [1.01;1.29]	M0	NA	1.11 [0.99;1.26] ‡	M0	NA	1.1 [0.98;1.25] ‡
Pancreas	M1b	1 month	1.18 [1.03;1.34]	M1	NA	1.07 [0.98;1.17]	M1	NA	1.07 [0.98;1.16]
	M1b	1 year	1.06 [0.94;1.19]						
	M1b	5 years	0.67 [0.41;1.08]						
Colon	M1	NA	1.21 [1.08;1.35]	M0	NA	1.05 [0.95;1.17] ‡	M0	NA	1.03 [0.93;1.13] ‡
Rectum	M1	NA	1.2 [1.07;1.34]	M0	NA	1.1 [0.98;1.24] ‡	M0	NA	1.07 [0.94;1.21] ‡
Bile ducts	M0	NA	NA	M0	NA	NA	M0	NA	NA
** *Females* **									
Stomach	M1b	1 month	1.48 [1.09;2.03]	M1b	1 month	1.48 [1.08;2.02]	M1b	1 month	1.47 [1.08;2.01]
	M1b	1 year	0.77 [0.57;1.05]	M1b	1 year	0.76 [0.55;1.03]	M1b	1 year	0.75 [0.55;1.02]
	M1b	5 years	0.49 [0.12;1.97]	M1b	5 years	0.44 [0.1;1.86]	M1b	5 years	0.43 [0.1;1.85]
Liver	M1	NA	1.18 [1.02;1.35]	M1	NA	1.17 [1.01;1.34]	M1	NA	1.16 [1.01;1.34]
Esophagus	M1	NA	1.44 [1.13;1.83]	M1	NA	1.41 [1.12;1.79]	M1	NA	1.41 [1.11;1.78]
Pancreas	M1	NA	1.08 [0.99;1.19]	M1	NA	1.08 [0.98;1.18]	M1b	1 year	1.03 [0.92;1.15]
							M1b	5 years	0.72 [0.39;1.3]
Colon	M1b	1 year	1.23 [1.06;1.43]	M0	NA	1.06 [0.97;1.17] ‡	M0	NA	1.05 [0.95;1.15] ‡
	M1b	5 years	0.61 [0.36;1.04]						
Rectum	M1	NA	1.23 [1.09;1.39]	M1	NA	1.18 [1.05;1.34]	M1	NA	1.17 [1.03;1.32]
Bile ducts	M1b	1 year	1.57 [1.21;2.02]	M1	NA	1.42 [1.14;1.76]	M1	NA	1.41 [1.13;1.75]
	M1b	5 years	2.06 [0.73;5.78]						
** *Males&Females* **									
Small intestine	M0	NA	NA	M0	NA	NA	M0	NA	NA

CI: confidence interval; EHR: excess mortality hazard ratios; mQ*i*: median value of national deprivation quintile *i* (see Section 2); NA: not applicable; SA: sensitivity analyses: (1) Simulated deprivation-specific French life tables derived from deprivation-specific England life tables; (2) Simulated deprivation-specific French life tables derived from mortality rate ratios by net income per consumption unit provided by The Permanent Demographic Sample (Echantillon Démographique Permanent, EDP). † Effect of EDI on excess mortality hazard; M0: not significant; M1: significant, steady over time since diagnosis and identical regardless of age at diagnosis; M1b: significant, time-dependent but not age-dependent; ‡ EHR estimated from model M1.

## Data Availability

Data and code are available upon reasonable request.

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
