# Peer review of "Socioeconomic Environment and Survival in Patients with Digestive Cancers: A French Population-Based Study"

_cancers, 2021, doi:10.3390/cancers13205156_

Round 1

Reviewer 1 Report

I would like to thank the authors and editors for giving me opportunity to read this paper. The authors raise a very relevant and interesting topic.

I have some comments on the chosen methods and on the results interpretation:

  1. In line 144 the authors write that the mortality hazard for cancer patients is a sum of excess mortality hazard and all cases of mortality hazard of the background population. They assume that the mortality hazard for cancer patients is always higher than all case mortality hazards of the background population, since they later toke a logarithm of excess mortality hazard. This assumption is not always true, a cancer population can be a specific subpopulation or a cancer treatment and can call special attention to that group of patients, and consequently they can be diagnosed earlier and treated for other diseases, which could give a higher mortality hazard rate.

  1. The authors claim that all case mortality hazards of the background population could differ according to socio-economic position and can be ignored referring to two sets of simulation. It is a crucial assumption for that paper and the performed simulations are not enough to convince the readers.

  1. The authors first chose a model using AIC and then estimated the chosen model, which is a two step procedure. Both the first and the second step have an uncertainty, but the shown results are not corrected for the uncertainty from the first step.   

  1. The interpretation of results is based on estimated confidence intervals, which are incorrect (see 3). The discussion is based on the presumption that the previously mentioned assumption is correct (see 1, 2). Moreover, throughout most of the discussion the effect of the disease stage is ignored, while it is well known that disease stage is one of the most important factors for cancer survival and that the social inequality has an influence on the disease stage at the time of diagnosis.

Reviewer 2 Report

I congratulate the authors on tackling an important and interesting question. However, I have some reservations regarding the approach and think that the manuscript should be improved. Authors should pay more attention to the fact that flexible modelling with multiplicity of outcomes and modelling choices likely produces lot of spurious findings. This perspective is insufficiently considered. A remedy is a more hypothesis driven approach and stronger consideration of randomness. Furthermore, cleared distinction of individual and regional inequality is necessary – in the abstract, introduction and discussion it should be always clear that the current findings are only based on social environment. It is not clear which of the previous findings were based on individual factors.

Detailed comments

  1. In the simple summary, it is not clear that social environment rather than individual social status is considered.
  2. Abstract lien 56: the verb “could” is not appropriate as there is lot of evidence and this evidence is even cited in the introduction.
  3. Introduction, line 78: I disagree that studying inequalities requires unbiased general population data and it is particular to the field of cancer.
  4. Line 85: The sentence “several reasons could explain…” is not accurately linked to the following sentences – these address specific aspects but not the social gradient
  5. Line 91 say the net survival was not studied – line 93 that often non-parametric net survival was used.
  6. Lines 92-94: The argument is made that non-parametric net survival analyses were not conducted in the past and therefore baseline hazard flexibility was not accounted for. This is not fully correct, actually non-parametric allows for flexibility. I agree that Cox regression does not display changes in hazard over time, but Kaplan-Meyer curves would display those – if they were applied to the specific question. So the point should be rather made that such questions were not asked at the content level. The technical argumentation based on application of a specific statistical method is insufficient – it should be rather explained why the authors expect that the social related survival difference might differ across time or age.
  7. In the introduction, it should be clearer what is known about individual poverty and regional poverty. The two levels are distinct and it should be clear what the manuscript is addressing (and what not). In the methods, it becomes clear that only regional inequality is considered.
  8. Results` section: I generally applaud the authors that they did not repeat lot of number in the text. On the other, side just naming tables content and not summarizing the findings in the text is insufficient. There is a plentitude of numbers and asking the reader to search for patterns on their own is moving responsibility. Clearly, the reader should be able to verify, but the authors should summarize the findings. To mention the content of the Figures and Tables is redundant because this one can see easily in the caption. So the text should be adapted to report results, not in which tables they are. (Results are summarized in the beginning of the discussion, but this is just an overview).
  9. In a more general perspective – the analysis considers many cancers and for each 4 models, stratified by sex – some of them show something, majority does not. Significance is not adjusted for multiple testing, so it is not surprising that some things appear. In the discussion, the authors refer to their findings as if they all were true – in fact, most likely several of the observations are random. Why digestive cancers survival varies (or not) with social gradient and why it depends on cancer site should be a background theory – against which the findings are discussed.
  10. A major limitation of this study is in my view that only regional inequality is considered. While regional inequalities can strongly affect access to care, individual factors can be closer to the outcomes related to behavior.

Reviewer 3 Report

The authors evaluate the impact of socioeconimic factors in the survival of patients with digestive cancer from France, as a means for improving and individualizing the management of these patients. Social and economic aspects are often overlooked in survival analysis of cancer patients, thus this study provides an important insight into the influence of these factors on survival. The study is well structured and thoroughly conducted. Social inequalities can be found worldwide and are a significant part in the care of cancer patients. The present study highlights the need for taking into account these inequalities in the diagnosis and treatment of cancer patients.

Reviewer 4 Report

The paper is of interest. Quality of the presentation must be improved. For example, I find tables too crowded and not easy to read. Please delete columns, keep only the essential to help the reader.

Why did you analyze only GI tumors? Is there a specific reason?

The period is really far from now (2006-2009). Don't you think this is a bias to your analysis? Maybe in more than 10 years conditions may be improved.

"We found that net survival was significantly worse for 62 individuals living in a more deprived environment as compared to those living in a less deprived one, for esophageal, liver, pancreatic, colon and rectal cancers, and for stomach and bile ducts cancers among females." Your conclusions are intuitive and appear obvious. Please speculate more, perhaps on the differences between different primary histologies.

Round 2

Reviewer 2 Report

The revision improved the manuscript and made it more balanced with respect to limitations of data and methods. I have no further comments.

Author Response

Thanks again for reviewing our manuscript.

(The reviewer did not ask for further revisions)

Reviewer 4 Report

The paper is suitable for publication

Author Response

(The authors gave the same response as above.)
